# Automatic Extraction of Power Lines from Aerial Images of Unmanned Aerial Vehicles

**DOI:** 10.3390/s22176431

**Published:** 2022-08-26

**Authors:** Jiang Song, Jianguo Qian, Yongrong Li, Zhengjun Liu, Yiming Chen, Jianchang Chen

**Affiliations:** 1Chinese Academy of Surveying & Mapping, Beijing 100036, China; 2School of Geomatics, Liaoning Technical University, Fuxin 123000, China; 3Faculty of Geomatics, Lanzhou Jiaotong University, Lanzhou 730070, China

**Keywords:** UAV, power line segmentation, aerial images, deep learning, Mask RCNN, connected domain grouping

## Abstract

Automatic power line extraction from aerial images of unmanned aerial vehicles is one of the key technologies of power line inspection. However, the faint power line targets and complex image backgrounds make the extraction of power lines a greater challenge. In this paper, a new power line extraction method is proposed, which has two innovative points. Innovation point one, based on the introduction of the Mask RCNN network algorithm, proposes a block extraction strategy to realize the preliminary extraction of power lines with the idea of “part first and then the whole”. This strategy globally reduces the anchor frame size, increases the proportion of power lines in the feature map, and reduces the accuracy degradation caused by the original negative anchor frames being misclassified as positive anchor frames. Innovation point two, the proposed connected domain group fitting algorithm solves the problem of broken and mis-extracted power lines even after the initial extraction and solves the problem of incomplete extraction of power lines by background texture interference. Through experiments on 60 images covering different complex image backgrounds, the performance of the proposed method far exceeds that of commonly used methods such as LSD, Yolact++, and Mask RCNN. DSC_PL_, TPR, precision, and accuracy are as high as 73.95, 81.75, 69.28, and 99.15, respectively, while FDR is only 30.72. The experimental results show that the proposed algorithm has good performance and can accomplish the task of power line extraction under complex image backgrounds. The algorithm in this paper solves the main problems of power line extraction and proves the feasibility of the algorithm in other scenarios. In the future, the dataset will be expanded to improve the performance of the algorithm in different scenarios.

## 1. Introduction

In recent years, the scale of China’s power grid has been rising and line complexity has been increasing, bringing huge challenges to grid inspection. Inspection work is extremely important to maintain the safe and stable operation of the regional power grid. Commonly used electric power inspection methods at home and abroad include the five methods of traditional manual inspection, manned helicopter inspection, robot inspection, satellite inspection, and UAV inspection. Among them, UAV inspection has the advantages of low cost and high efficiency compared with other methods and has been widely used.

The accurate extraction of power lines is one of the core steps of UAV inspection [1,2] and, as the basis for distance monitoring [3,4,5], broken strand monitoring [6,7,8], ice cover monitoring [9,10,11], foreign object monitoring [12,13], and power line arc sag measurement [14,15,16] of hazardous operations, it is important to extract power lines accurately and quickly. Laser point cloud data have high accuracy and high-density 3D spatial information [17,18,19,20] and are used by some scholars for fine power line extraction; the literature [21] realizes single power line fine extraction based on the residual clustering method. However, this type of method is costly and not suitable for large-scale power rapid inspection. In contrast, visible image inspection is low cost, has intuitive data, and has high engineering application value for power line inspection [22,23]. The current research on visible image power line extraction can be broadly divided into three categories: power line extraction based on edge detection operator, power line extraction based on joint features, and power line extraction based on deep learning.

Edge detection operator-based power line extraction methods are divided into two categories based on whether a priori knowledge is introduced or not. One class combines the brightness and the direction of the power lines as a priori knowledge with the edge detection operator for power line extraction [24,25,26,27,28,29]; the other class extracts the power lines in the edge map directly after the edge detection operator acquires the edge map using the Hough transform (Hough) [30,31]. Although these methods have a simple structure, they ignore the problem of interference noise suppression. The joint feature-based power line extraction algorithm uses objects that coexist with power lines, such as towers and insulators, as auxiliary features for power line extraction. The literature [32] defines line–tower spatial association features based on the spatial relationship between power lines and towers and combines the line features with spatial contextual information in the area around the lines to achieve power line extraction. The literature [33] uses the corner points of electric poles as the height basis for line extraction to extract power lines. This type of algorithm has a large limitation, the algorithm generalization ability is limited, and the algorithm performance decreases rapidly when the test image does not match the predefined association model.

With the great success of deep learning in the fields of image classification and image segmentation [34,35], more and more scholars have started to focus on the research of implementing visible image power line extraction based on deep learning [36]. The literature [37] used a convolutional neural network (CNN) to classify the edge detector detection map and to perform power line fitting by Hough transform. The literature [38] extracted features using CNN and used a PCA (principal component analysis) algorithm for dimensionality reduction and classification by a generic machine learning algorithm. The literature [39] proposed a fast power line detection network (fast PLDN) that efficiently extracts the spatial and semantic information of power lines using low high-pass filters and edge attention fusion modules. However, the literature [37,38] can only achieve to determine whether the image block contains power lines and cannot accurately locate the power lines in the graph. The literature [39] does not effectively resolve interference from other linear targets and exhibits less than optimal performance for closely spaced adjacent power lines. Jayavardhana et al. [40] proposed a classifier to determine “Line present” and “No line present”. Van Nhan Nguyen et al. [36] proposed a fully convolutional feature extractor, classifier, and line segment regressor to achieve a real-time detection of power lines with the VGG-16 network as the backbone. However, these methods obtain some false positive pixels when extracting power lines, which make the power lines thicker or longer.

In addition, the following characteristics of power lines make it difficult to extract power lines from UAV aerial images based on deep learning. First, power lines are faint targets in aerial images, with a small relative pixel share. Secondly, power lines often pass across or vertically through the whole image. Third, the power lines have a thin and long physical structure, especially in the vertical power line direction, which accounts for only a few pixels in width. Fourth, the power corridor environment is complex and the image background is variable, often accompanied by interference targets with similar characteristics to the power lines.

To address the above problems, a new automatic power line extraction algorithm is proposed in this paper. The algorithm uses Mask RCNN [41], which has the top segmentation performance at present, as a deep learning network framework, adopts the strategy of block processing for the initial extraction of power lines, and uses the connected domain group fitting algorithm to solve the problems of power line breakage and mis-extraction after initial extraction of power lines. The algorithm has the following advantages.

(1)Using the block processing strategy, the anchor frame size is reduced globally to increase the proportion of power lines in the feature map and to reduce the accuracy degradation caused by the original negative anchor frames being misclassified as positive anchor frames.(2)Further processing of the initially extracted power lines uses the connected domain group fitting algorithm to solve the problem of power line breakage and mis-extraction.(3)Compared with the traditional Mask RCNN method, the extraction accuracy, precision, and anti-interference performance of the algorithm in this paper are greatly improved.

## 2. Related Work

### 2.1. Data Acquisition

The measurement area was selected in the line from tower 09 to tower 37 of Yangu Line in Daling Village, Dapu Town, Hengdong County, Hunan Province, China, adjacent to the village with no tall buildings around. As shown in Figure 1, the measurement area covers a variety of land types such as forests, grasslands, cultivated land, roads, houses and ponds, and water, reflecting more comprehensively the complex background of the UAV aerial photography power inspection images.

When collecting data, the drone flew directly above the power line, setting the heading overlap rate to 70% and the cruising speed to 50 km/h. It flew in one direction from pole tower 09 to pole tower 37, shooting every 5 s.

### 2.2. Operation Equipment

The DJI Genie Phantom 4RTK UAV is a small multi-rotor high-precision aerial survey drone for low-altitude photogrammetry applications with a centimeter level navigation and positioning system and a high-performance imaging system that is portable and easy to use. The DJI Genie Phantom 4RTK consists of an aerial vehicle, a remote control, a gimbal camera, and the accompanying DJIGS RTK App. The aircraft parameters are shown in Table 1.

### 2.3. Dataset Production

In this paper, we use a chunking strategy to crop the images into suitable sizes before inputting them into the model, so when we make the dataset we also crop the images into suitable sizes in advance. As shown in Figure 2, the original size of 3840 × 2160 images is cut into 4 × 4 equal parts, each with the size of 960 × 540, and the images containing power lines are selected for manual labeling. In this paper, we use the Labelme annotation tool to completely fit the powerline contours and we use the polygon annotation method to annotate the powerline contours in the inspection images, which is called powerline. A total of 300 images containing powerlines are selected for annotation in this dataset, and the size of each image is 960 × 540, which is cut into the training and test sets according to the ratio of 7:3.

## 3. Methodology

In this section, we will describe our proposed method in detail; however, before that we will first briefly describe the process of this method. The structure of the proposed method is shown in Figure 3. It mainly includes four steps: (1) first, we will segment the image extracted from the UAV Power Patrol video and number each block; (2) second, each block is sequentially input into the Mask RCNN model to extract power lines and then spliced to obtain power line masks; (3) finally, a connected domain grouping fitting algorithm is proposed to further process the initially extracted power lines to obtain a complete power line.

### 3.1. Power Patrol Image Chunking

The size of the image extracted from the UAV Power Patrol video is large and the power line accounts for less in the image due to its slender physical structure. Often, a single power line accounts for only about 0.1% of the image, which brings great difficulties to the subsequent power line extraction tasks. In addition, the power line often crosses or vertically crosses the whole image, resulting in a high probability of misjudging positive samples in the RPN (risk priority number) [42,43] link of Mask RCNN. Based on this, the algorithm adopts a block processing strategy, which reduces the size of the anchor frame globally to improve the proportion of power lines in the feature map. At the same time, the strategy can also reduce the misjudgment of negative samples as positive samples in RPN to improve the accuracy of power line extraction.

Firstly, the UAV power inspection video is a frame extracted according to the overlap of 60%, and the power inspection image with the size of hxw is obtained. The image is cut into nxm copies and each copy is numbered. If the edge of the image is not enough to be divided into one, this part is filled with 0.

### 3.2. Mask RCNN Preliminary Extraction of Power Line

Mask RCNN is an instance segmentation network model developed by He et al., based on Faster RCNN [44], which can accomplish the task of semantic segmentation with high accuracy while accomplishing target recognition; it is one of the best current techniques in object recognition and object segmentation [45,46]. The algorithm uses a residual network (ResNet) [47,48,49,50] as the backbone network and extracts feature elements at multiple scales by constructing top-down feature pyramid networks (FPN) [51,52,53]. Regions of interest (RoI) are selected using region proposal networks (RPN), and a fixed-size feature map is generated using the RoI Align method and input into the fully connected network. Finally, fully connected classification, bounding box regression, and mask regression are used to achieve instance segmentation of the target, and the structure of the Mask RCNN network is shown in Figure 4.

The segmented local power inspection images are successively input into Mask RCNN for power line extraction. The key link for Mask RCNN to extract power lines is RPN. This link extracts the negative sample area containing the background and the positive sample area containing the target information from the image and inputs the same amount of positive and negative samples into the subsequent network to ensure the balance of positive and negative samples in the data. Usually, the IoU (intersection over union) of the prediction box and the truth box is calculated to judge the positive and negative samples according to the threshold set in advance. Because the power line has a thin and long physical structure and often crosses or crosses the whole image, the problem of framing the prediction that meets the IoU threshold but does not include the power line into positive samples often occurs in the extraction of positive samples. As shown in Figure 5, A is the truth box, B is the prediction box, and the IoU calculation formula of A and B is:(1)$${\mathrm{IoU}}_{\mathrm{AB}}=\frac{A\cap B}{A\cup B}$$

At this time, the IoU of A and B satisfy the threshold, but the prediction frame B does not contain power lines, and such cases will be misclassified as positive samples in the RPN session. To reduce the occurrence of this type of problem, this paper adopts a chunking strategy to reduce the anchor box size globally. When the anchor frame size is reduced, the true value frame A will be significantly reduced, the proportion of power lines in the true value frame will be increased, and the intersection of the prediction frame B and the true value frame A without power lines will also be significantly reduced, which largely avoids the occurrence of positive sample misclassification in the RPN session and improves the accuracy of Mask RCNN in extracting power lines.

At the same time, the strategy of chunking can increase the pixel share of power lines in the map and reduce the difficulty of power line extraction. When the original image size of the aerial image is 3840 × 2160, the pixel share of a single power line is less than 0.1% and the target is extremely weak, which makes the feature extraction work more difficult. When the original image is divided into 4 × 4 chunks, the image size is 960 × 540 and the pixel share of a single power line reaches 0.5%, which is about 5 times higher and greatly reduces the difficulty of power line extraction.

The power patrol image is chunked and input into Mask RCNN to extract power lines, obtain local power line masks, stitch the local power line masks according to the number in turn, remove the filled part, and finally obtain the original image size power lines.

### 3.3. Connected Domain Group Fitting Algorithm

The power lines extracted by Mask RCNN inevitably have breakages and mis-extractions, which need to be further processed. The Hough transform [54,55,56,57], as a classical linear detection algorithm, is often used for linear object detection. However, the time complexity and space complexity of this algorithm is high and the results are often less than ideal. To address these problems, the connected domain group fitting algorithm (CDGFA) is proposed in this paper. The power lines are grouped using the connected domain, and the least squares algorithm is used to fit the power lines within the connected domain component. Finally, multiple line segments on the same power line are connected according to the feature of basic parallelism between power lines, and the mis-extracted targets are eliminated. The algorithm flow is shown in Figure 6.

Step 1: Perform morphological corrosion operation and morphological expansion operation [58,59,60] on Mask RCNN extraction results to eliminate noise points and bridge small gaps.

Step 2: connected domain analysis [61,62] is performed on all power lines and labels are attached. It is roughly divided into two scans. First scan: For each foreground pixel encountered, determine whether the surrounding pixels already have labels and, if so, leave them unchanged, if not, tag new labels. If there are two labels adjacent to each other, the adjacency relationship is recorded. The second scan: when a foreground pixel with a label is encountered, it is relabeled according to the adjacency relationship until each connected domain has its label. As shown in Figure 7, each connected domain has its label, and each color represents one label.

Step 3: The slope and length of each connected domain are calculated by the least squares algorithm to fit the power line. The equation of the fitted straight line for the connected domain is that the difference between the fitted value and the sample value is the error, and the sum of squares of the error measures the total error.
(2)$$J(a,b)=\sum_{k=1}^{m}{\left(y_{k}-ax_{k}-b\right)}^{2}$$

For the error function, there is an extreme value when its derivative is 0. Therefore, find the partial derivative of the error function and make it 0.
(3)$$\begin{array}{l} \frac{\partial J}{\partial a}=\sum_{k=1}^{m}-2x_{k}(y_{k}-ax_{k}-b)=0\\ \frac{\partial J}{\partial b}=\sum_{k=1}^{m}-2(y_{k}-ax_{k}-b)=0 \end{array}$$

Appointment $S(xy)=\sum x_{k}y_{k}$, $S^{2}(x^{2})=\sum x^{2}{}_{k}$, $S(x)=\sum x_{k}$, $E(x)=\frac{1}{m}\sum x_{k}$, $E(y)=\frac{1}{m}\sum y_{k}$. Then we can obtain Equation (4):(4)$$\begin{array}{l} a=\frac{S(xy)-E(y)S(x)}{S(x^{2})-E(x)S(x)}\\ b=\frac{E(y)S(x^{2})-S(xy)E(x)}{S(x^{2})-E(x)S(x)} \end{array}$$

The slope *k* and area *s* of each connected domain are taken out by the connected domain label, where the connected domain with the largest area L, the slope of that connected domain is *K*. If the *i*-th connected domain satisfies $|K-k_{i}|>tan\;5$ or $s_{i}<500$, eliminate this connectivity domain. This step eliminates the connected domains that do not contain power lines.

Step 4: Group the connected domains and fit each group of power lines. The distance from the midpoint of each connected domain to the largest connected domain L in the area is calculated and added to the set *D*. The set *D* is sorted in ascending order. Subsequently, the connected domains are grouped according to set *D*. When the neighboring distance connected domains in the set *D* satisfy $|d_{i+1}-d_{i}|<d_{th}$, divide them into the same group. Finally, the coordinates of all the pixel points in the connected domain of each group are taken out using the connected domain labels, and the power lines of each group are fitted based on the least squares algorithm based on these coordinate points.

## 4. Experimental Results

### 4.1. Model Training

The power line automatic extraction algorithm model is built according to the method proposed in this paper and the self-built power line dataset is converted to COCO to train the network with the required dataset. The pre-training file uses mask_rcnn_R_50_FPN_3x, the ResNet50/101+FPN model is used as the backbone network, and the hyperparameters of the model are obtained by genetic algorithm. The initial hyperparameters are shown in Table 2.

### 4.2. Experimental Data

Sixty images from three different lines were selected from the UAV power inspection videos to create a test dataset, as shown in Figure 8. These 60 images are all 3820 × 2160 in size and include a variety of complex image backgrounds, including water, woodland, and grass, which have a large color contrast with the power lines, as well as arable land, which has little color difference from the power lines. In addition, there are images with a large number of linear features such as roads and houses as backgrounds. These image backgrounds in each scene reflect the main challenges of power line extraction: (1) faint power line targets; (2) power line colors similar to image backgrounds and power lines blending into image backgrounds; (3) interference of linear features. If the algorithm performs well on this test dataset, it proves that the algorithm can solve the main challenges of power line extraction and has the potential to extract power lines in various complex scenes.

### 4.3. Evaluation Parameters

Almost all studies on powerline detection use dice score (DSC) (also known as F-score) [63], precision, true positive rate (TPR) (also known as recall or sensitivity), false discovery rate (FDR), and accuracy. These assessment parameters are defined as:DSC or F-score = 2TP/(2TP + FP +FN)(5)
Precision = TP/(TP + FP)(6)
TPR or Recall or Sensitivity = TP/(TP + FN)(7)
FDR = FP/(FP + TP)(8)
Accuracy = (TP + TN)/(TP + TN + FP + FN)(9)
where TP denotes the number of pixels with both predicted and actual power lines; TN denotes the number of pixels with neither predicted nor actual power lines; FP denotes the number of pixels with predicted power lines and actual background; FN denotes the number of pixels with predicted background and actual power lines. Accuracy refers to the ratio of the number of correctly predicted samples to the total number of predicted samples and it does not consider whether the predicted samples are positive or negative. Precision refers to the ratio of the number of correctly predicted positive samples to the number of all predicted positive samples, i.e., how many of all predicted positive samples are positive samples. The recall is the ratio of the number of correctly predicted positive samples to the total number of true positive samples, i.e., how many positive samples can be correctly identified from these samples. FDR refers to the percentage of failure detection in the foreground power line class.

### 4.4. Comparison with Other Methods

In this paper, three methods are chosen to compare with our approach. The first one is the Line Segment Detector (LSD) [64,65,66], which is a classical line segment detection method that detects line segments based on the change of pixel gradient; therefore, it is often used for linear target detection tasks, such as detecting power lines. The second method is Yolact++ [67], which belongs to the one-stage model and is faster compared with Mask RCNN, but slightly less accurate. The literature [68] tested the self-built transmission tower and power line dataset (TTPLA) on its basic version (Yolact [69]) and achieved good results. The third method is Mask RCNN [70,71], which belongs to the two-stage model and is currently one of the best methods for instance segmentation accuracy.

These methods are highly feasible for achieving the power line detection task and are some of the most advanced methods available, and the comparison of the method in this paper with these methods can prove the advanced nature of the method in this paper. When we use the Yolact++ method, we choose Resnet101-FPN, which has the best segmentation accuracy, as the backbone network. Because the image size has a large impact on the segmentation performance of the method, we found after experimental tests that the method performs best on our experimental data when the image size is 480 × 270. To facilitate highlighting the effectiveness of the chunking strategy proposed by the method in this paper, the original image size (3840 × 2160) is used in this paper when using the Mask RCNN method.

Table 3 shows the results of comparing the five performance metrics of these three methods with the method of this paper, namely dice score (DSC), true positive rate (TPR), false discovery rate (FDR), and precision (Precision), and accuracy (Accuracy).

As can be seen from the results in Table 3 and Figure 9, LSD is sensitive to linear features and can accurately detect power lines in various scenes. However, the method cannot effectively determine power lines and non-power lines and contains a large number of incorrectly detected targets in the detection results. In the five sets of scene detection results in Figure 9, LSD detects most of the power lines, but unfortunately a large number of non-power lines in the image background are also extracted. For example, the electric towers in groups (a) and (b) and the houses in groups (c), (d), and (e) are detected together. Compared with LSD, Yolact++ performs much better. In Table 3, the TPR of this method reaches 82.10, which is the highest value among the four methods, indicating that Yolact++ is able to detect power lines well from various image backgrounds. However, the FDR value of this method is as high as 64.79, which is also the maximum among the four methods, indicating that the mis-extracted rate of this method is also relatively high. Specifically, in the five sets of results in Figure 9, Yolact++ detects most of the power lines, but the features similar to power lines in various scenes, such as houses and roads, bring greater interference to the power line extraction task. In particular, a large number of mis-extracted results are included in the detection results in the group (c). In this experiment, the training dataset is only about 200 images and the amount of data is relatively small, resulting in less mature training of the Mask RCNN network model, so the results of this method are not satisfactory in this experiment. If the training dataset is supplemented, the performance of the method has room for further improvement.

In contrast, the method in this paper performs very well in all aspects. The DSC_PL_, precision, and accuracy of the method are 73.95, 69.28, and 99.15, respectively, which are the highest values, and the FDR is only 30.72, which is the lowest value. In addition, the difference between the TPR value and the highest value is only 0.35. This indicates that the method in this paper not only extracts power lines accurately and correctly but also avoids most mis-extracted cases. Specifically, in the five sets of results in Figure 9, we can know that the method in this paper can extract every power line almost completely, no matter whether it is a complex scene such as (c) or a simple scene such as (e), no matter whether it is (b) where the power lines are contrasted with the image background or (a) where the power lines are integrated into the image background. Although there are inevitably a small number of broken power lines and mis-extractions, the performance demonstrated by the method in this paper is sufficient for practical engineering needs.

## 5. Discussions

### 5.1. Effectiveness of Chunking Strategies and the Impact of Chunk Size on Performance

The algorithm in this paper implements the initial extraction task of power lines with the Mask RCNN network framework. However, the power lines are weakly targeted in the images, which makes the extraction task more difficult. Based on this, this paper proposes a chunking extraction strategy to solve the problem. To demonstrate the effectiveness of the chunking strategy and to investigate the effect of chunk size on the performance of the algorithm, we crop 60 images in the test dataset into 480 × 270, 960 × 540, 1920 × 1080, and 3840 × 2160 sizes, respectively, into Mask RCNN for the power line extraction task. Table 4 shows the average values of each performance parameter under different chunk sizes.

According to Table 4 and Figure 10, we can know that the algorithm exhibits different performances when the image block size varies. When the chunk size is 3840 × 2160, Mask RCNN has the worst performance parameters in each aspect and the most drastic fluctuations in each performance parameter on the test dataset. When the chunk size is 480 × 270 and 1920 × 1080, the performance of Mask RCNN improves and the trend of the variation of the performance parameters is relatively stable. When the chunk size is 960 × 540, Mask RCNN achieves the strongest comprehensive performance with DSC_PL_, TPR, and accuracy of 63.26, 86.31, and 98.94, respectively, which are the highest values. Moreover, the FDR and accuracy are similar for this size compared with the 1920 × 1080 data block size. Therefore, this indicates that Mask RCNN achieves the best performance on this test dataset when the data block size is 960 × 540.

The performance comparison of Mask RCNN at chunk size 960 × 540 and the original image size 3840 × 2160 shows that DSC_PL_, TPR, precision, and accuracy are improved by 16.10, 29.26, 7.90, and 2.13, respectively, and FDR is reduced by 6.23. It can be seen that Mask RCNN at chunk size has a significant improvement in power line extraction performance under the chunk size strategy, which can illustrate the effectiveness of the chunking strategy proposed in this paper.

Specifically, as the two sets of extraction results in Figure 11 show, when the chunk size is 3840 × 2160, Mask RCNN cannot extract power lines with a similar color to the image background, and the power lines extracted at this site are accompanied by a large number of false positive pixels, resulting in thicker power lines. When the chunk size is 1920 × 1080 and 480 × 270, Mask RCNN extracts power lines much better and is able to extract most of the power lines. However, there are still some broken and incomplete extraction cases. When the chunk size is 960 × 540, Mask RCNN has the best performance in extracting power lines and can basically extract the power lines completely. This shows the effectiveness of the proposed chunking strategy in helping Mask RCNN to extract power lines, and the best performance of Mask RCNN in extracting power lines on this test dataset is demonstrated by experiments when the chunk size is 960 × 540.

### 5.2. Performance of the Connected Domain Group Fitting Algorithm

After the initial extraction of power lines by Mask RCNN, there are inevitably some broken and misleading cases, which cause greater interference to the power line extraction work. Based on this, a connected domain group fitting algorithm is proposed in this paper to further process the initially extracted power lines and to achieve complete and accurate extraction of power lines in complex image backgrounds by the algorithm in this paper. We input the 60 images in the test dataset with a block size of 960 × 540 into Mask RCNN, and Mask RCNN combined with the connected domain group fitting algorithm to obtain the algorithm in this paper; the extraction results obtained are shown in Table 5.

According to Table 5, we can see that the performance of the mask RCNN is significantly improved by adding the connection domain group fitting algorithm. The DSC_PL_, accuracy, and precision are increased by 10.69, 18.33, and 0.21, respectively, and the FDR is reduced by 18.33. In addition, although the TPR is smaller than the mask RCNN method, the difference is only 4.56. It can be seen that the connection domain proposed in this paper group fitting algorithm can effectively solve the fault and mis-extracted problems after the extraction of power lines by Mask RCNN and can effectively improve the performance of this algorithm.

Figure 12 shows the comparison of the two methods to solve the power line breakage problem. Comparing the results of the two methods for extracting power lines, it can be seen that the Mask RCNN method cannot solve the power line breakage problem, while the method in this paper can solve this problem. This indicates that the connected domain group fitting algorithm in this paper can solve the power line breaking problem after the initial extraction of power lines.

Figure 13 shows the comparison of two methods to solve the power line error detection problem. From the figure, we can know that there are two error detections in the extraction result of Mask RCNN. In the method of this paper, these two error detections are eliminated. This indicates that the connected group fitting algorithm in this paper can solve the mis-extracted problem after the initial extraction of power lines.

In summary, we can know that the connected domain grouping fitting algorithm in this paper can effectively solve the power line breakage and mis-extracted problems that exist after the initial extraction of power lines and can improve the performance of the algorithm in this paper. From the experiments, we can also know that the performance of the method in this paper has a great advantage over other classical power line extraction methods [64,68,70] such as the traditional Mask RCNN.

## 6. Conclusions

In this paper, we propose a new automatic power line extraction algorithm, which is divided into two parts: the initial extraction of power lines based on Mask RCNN using a chunk extraction strategy; the further processing of power lines using a connected domain group fitting algorithm to extract the complete power lines. The algorithm in this paper mainly solves the following problems:(1)To address the problems of power lines running through the whole map and the difficulty of extraction due to the faint target, this paper adopts a chunking extraction strategy to globally reduce the anchor frame size to increase the proportion of power lines in the feature map. In addition, this strategy can reduce the accuracy degradation caused by the original negative anchor frame being misclassified as a positive anchor frame.(2)The proposed connected group fitting algorithm can effectively solve the problems of breakage and mis-extraction after the initial extraction of power lines.

Three commonly used methods, LSD, Yolact++, and Mask RCNN, are used to conduct comparison experiments with this paper’s method based on 60 test datasets of power lines covering a variety of scenes. The performance parameters of the algorithm in this paper are optimal except for TPR which is 81.75, only 0.35 from the highest value, DSCPL, precision, and accuracy are 73.95, 69.28, and 99.15, respectively, and FDR is only 30.72. The experimental results show that the algorithm in this paper has excellent performance, can solve the main problems of extracting power lines, and can realize the task of power line extraction in the complex image background. The experimental results show that this algorithm has excellent performance and can solve the main challenges of power line extraction. In addition, two innovations proposed in this paper, the chunk extraction strategy and the connected domain group fitting algorithm, have also proved their effectiveness through experiments.

In conclusion, the algorithm in this paper can solve the main problems of power line extraction in UAV aerial images and the algorithm has superior performance and high engineering application value.

## Figures and Tables

**Figure 1 sensors-22-06431-f001:**
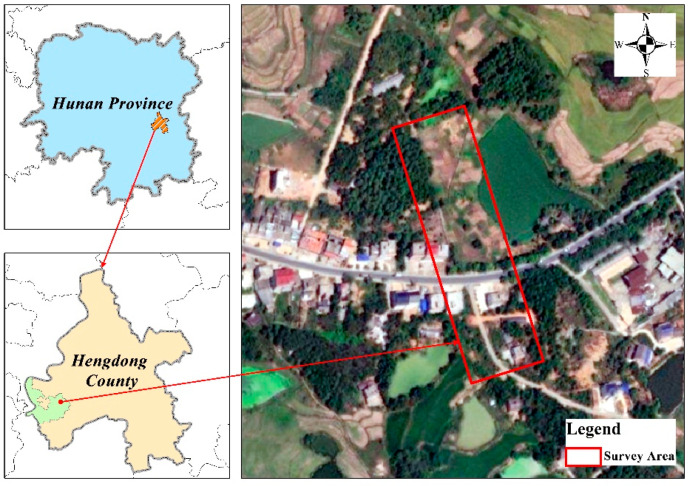
Overview of survey area.

**Figure 2 sensors-22-06431-f002:**
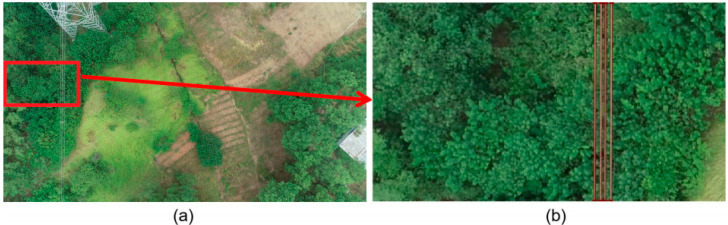
Data set making. (**a**) is an aerial drone image of the power lines; (**b**) is a partial image of (**a**) with labels added to the power lines in the image.

**Figure 3 sensors-22-06431-f003:**
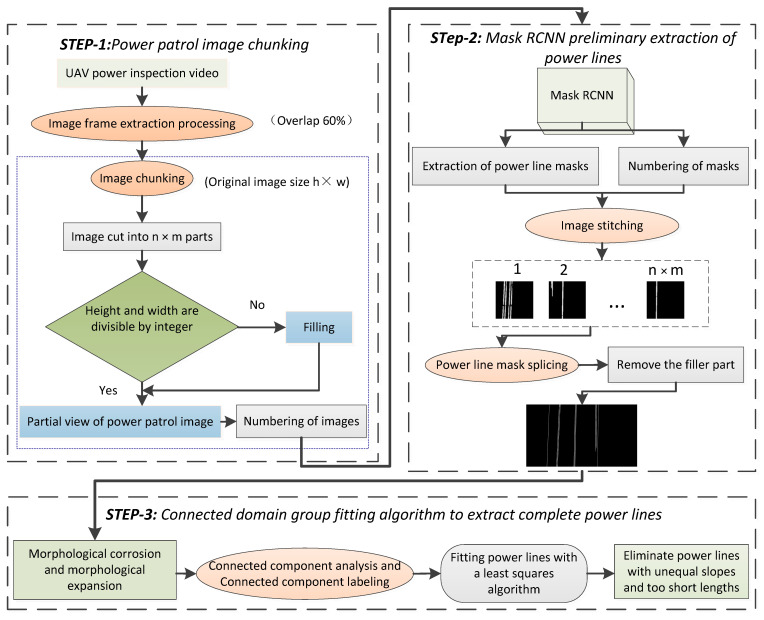
Automatic extraction technology of power lines from UAV aerial images.

**Figure 4 sensors-22-06431-f004:**
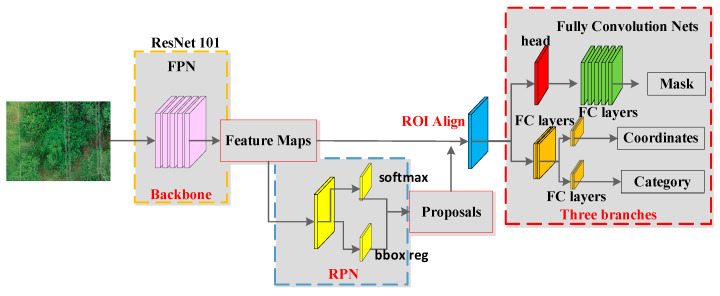
Mask RCNN network main structure diagram.

**Figure 5 sensors-22-06431-f005:**
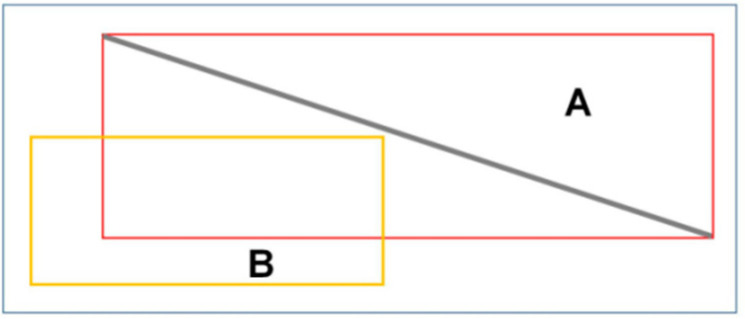
Positive sample misclassification cases. A is the truth box, B is the prediction box.

**Figure 6 sensors-22-06431-f006:**
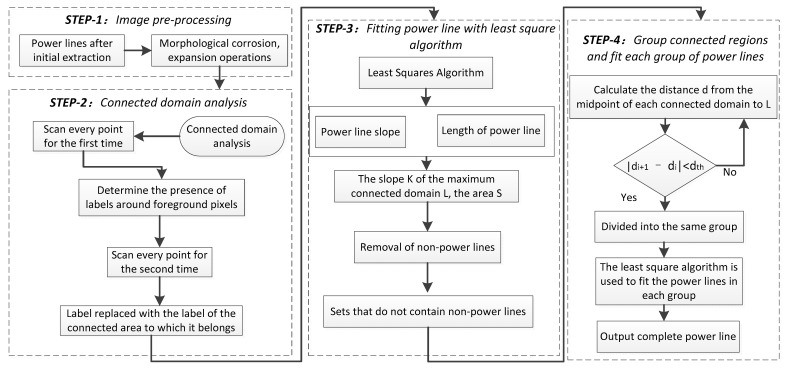
Connected domain group fitting algorithm.

**Figure 7 sensors-22-06431-f007:**
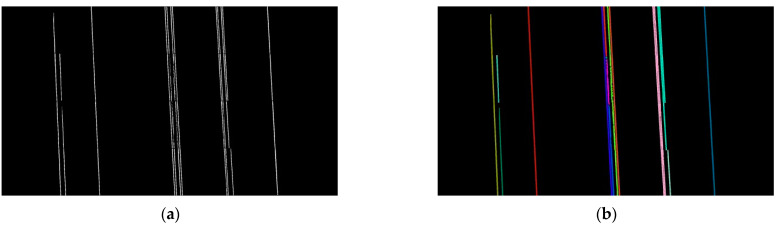
Attach different labels to each connected domain. (**a**) Before adding tags; (**b**) after adding tags.

**Figure 8 sensors-22-06431-f008:**
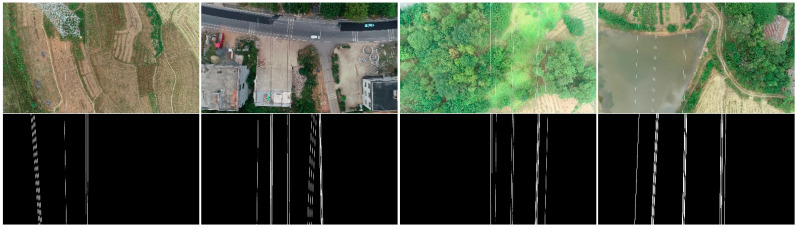
Partial image of the test dataset.

**Figure 9 sensors-22-06431-f009:**
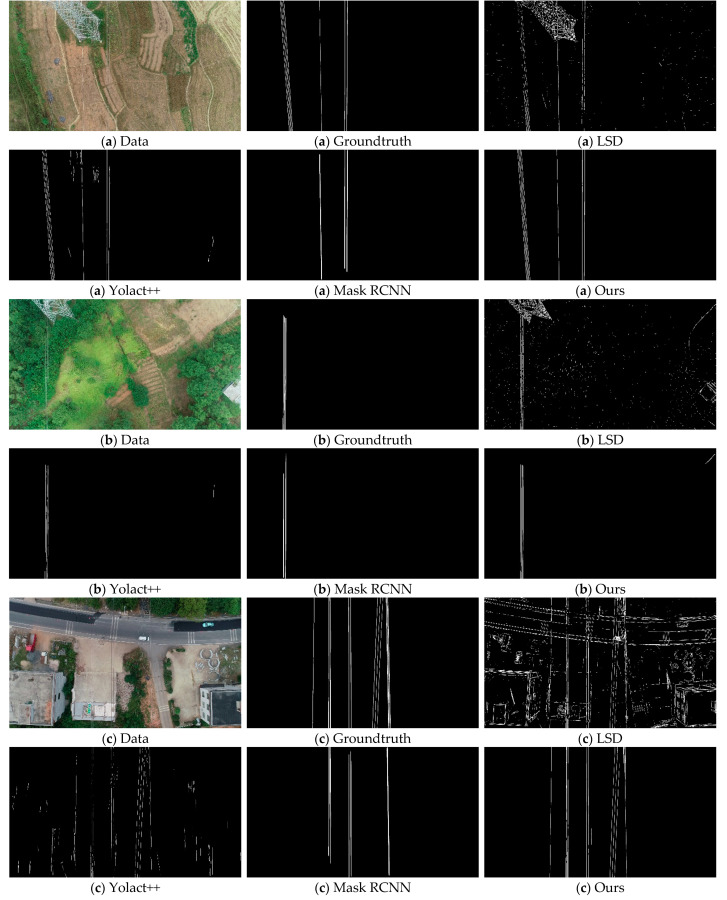
The extraction results of various methods on the test dataset. The extraction results are shown for five scenes, namely (**a**–**e**). A total of six images are shown for each scene, which are the input image (Data), the truth image (Groundtruth), and the extraction results of four methods (LSD, Yolact++, Mask RCNN, and Ours).

**Figure 10 sensors-22-06431-f010:**
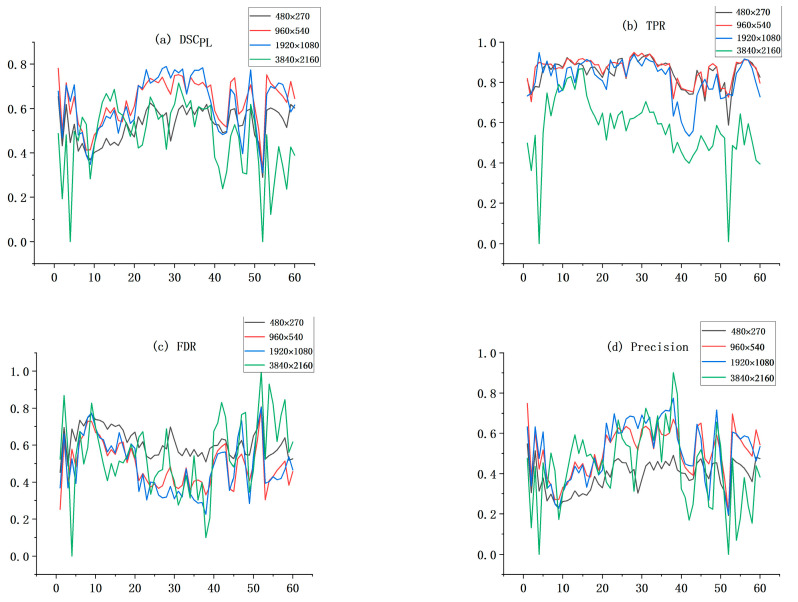
The trend of each performance parameter of Mask RCNN in the test dataset with different chunk sizes. (**a**–**e**) is the trend plots of the performance parameters (DSCPL, TPR, FDR, Precision, and Accuracy) at each size over 60 images, respectively, and (**f**) is the trend plot of the mean values of these five parameters at different sizes.

**Figure 11 sensors-22-06431-f011:**
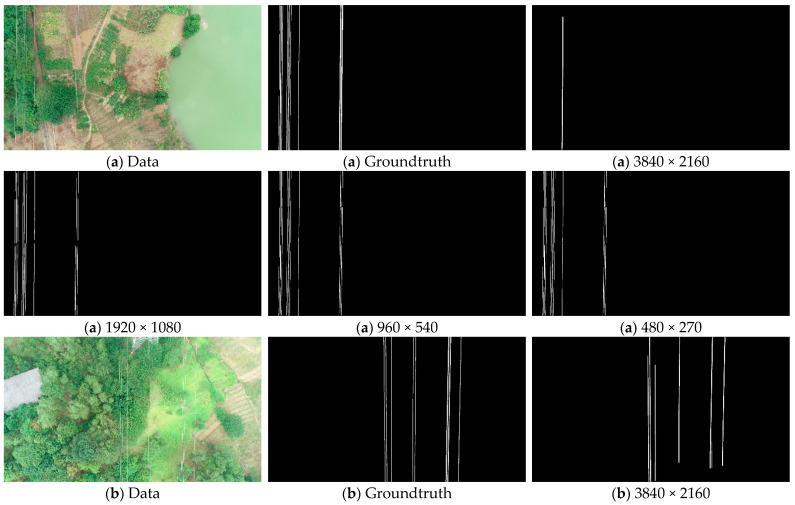
Extraction results of Mask RCNN at different sizes. There are (**a**,**b**) two groups of extraction results with different image backgrounds, each group has six images, which are the input image (Data), the truth image (Groundtruth), and the extraction results in four sizes (480 × 270, 960 × 540, 1920 × 1080, and 3840 × 2160).

**Figure 12 sensors-22-06431-f012:**
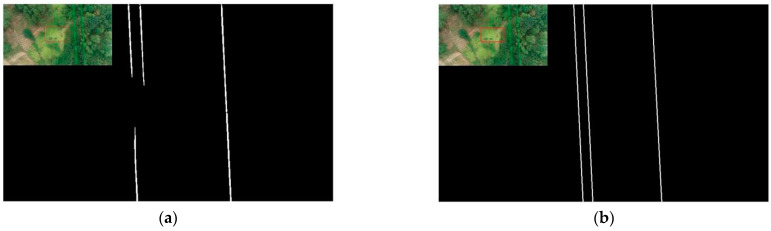
Comparison of two methods for solving the power line breakage problem. (**a**) Mask RCNN; (**b**) Mask RCNN + CDGFA.

**Figure 13 sensors-22-06431-f013:**
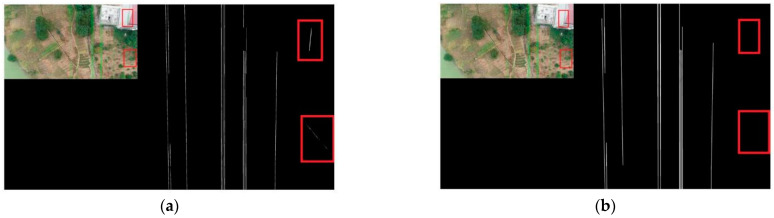
Comparison of two methods to solve the error detection problem. (**a**) Mask RCNN; (**b**) Mask RCNN + CDGFA.

**Table 1 sensors-22-06431-t001:** UAV parameters.

Camera GNSS	Parameters
Image sensor	inch CMOS; 20.0 million effective pixels (204.8 million total pixels)
Video resolution	H.264, 4 K: 3840 × 2160 30 p
Maximum photo resolution	4864 × 3648 (4:3)
Frequency of use	GPS: L1/L2 GLONASS: L1/L2 BeiDou: B1/B2 Galileo: E1/E5
Positioning accuracy	Vertical 1.5 cm + 1 ppm (RMS) Horizontal 1 cm + 1 ppm (RMS)

**Table 2 sensors-22-06431-t002:** Model training parameters.

Parameter	Value
weight decay	0.0001
learning rate	0.001
maximum iteration	36,000
ims_per_batch	4
batch_size_per_image	128

**Table 3 sensors-22-06431-t003:** Comparison of the performance of each of the four methods on the test dataset.

Method	DSC_PL_ (%)	TPR (%)	FDR (%)	Precision (%)	Accuracy (%)
LSD	49.74	52.90	49.26	50.74	98.20
Yolact++	48.56	**82.10**	64.79	35.21	98.66
Mask RCNN	47.16	57.05	55.28	43.05	96.81
Ours	**73.95**	81.75	**30.72**	**69.28**	**99.15**

**Table 4 sensors-22-06431-t004:** Based on 60 images in the test dataset, the average of each performance parameter of Mask RCNN at different chunk sizes.

Chunk Size	DSC_PL_ (%)	TPR (%)	FDR (%)	Precision (%)	Accuracy (%)
480 × 270	52.63	85.19	61.38	38.62	98.74
960 × 540	**63.26**	**86.31**	49.05	50.95	**98.94**
1920 × 1080	61.92	81.56	**48.37**	**51.63**	98.88
3840 × 2160	47.16	57.05	55.28	43.05	96.81

**Table 5 sensors-22-06431-t005:** The average of the performance parameters of the Mask RCNN and Mask RCNN + CDGFA extraction results when the block size of the test dataset is 960 × 540.

Method	DSC_PL_ (%)	TPR (%)	FDR (%)	Precision (%)	Accuracy (%)
Mask RCNN	63.26	**86.31**	49.05	50.95	98.94
Mask RCNN + CDGFA	**73.95**	81.75	**30.72**	**69.28**	**99.15**

## Data Availability

The data in this study are owned by the research group and will not be transmitted.

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
