# Peer review of "Automatic Extraction of Power Lines from Aerial Images of Unmanned Aerial Vehicles"

_sensors, 2022, doi:10.3390/s22176431_

Round 1

Reviewer 1 Report

The paper describes a Mask RCNN based power line extraction method from aerial images. Although the problem is challenging, many existing literatures are available. The proposed method produces fairly good detection results (average precision 58.89), but only tested on 6 images, of which the power lines are mostly over green vegetation backgrounds.

The followings are the main weaknesses of the paper:

1. If the power line extraction method is for inspection purposes, images of power line of different conditions (e.g. snow covered, broken strand. foreign objects) should be included.

2. No comparison with existing power line extraction work (reference 38 to 40). As a result, it is hard to judging whether a precision of 58.89 is good or not. Although a comparison with results from the classical Mask RCNN is included, it is not designed for detecting power lines so its performance is expected to be poorer.

As for the evaluation metric, precision would be the most important criteria  if this is for inspection purpose. As false positive pixel would make the line look incorrectly thicker or longer, while false negative pixel would have the opposite effect. You should make it clear what the requirements are and choose the evaluation metric appropriately. I don't think you need to consider correct detection of background pixels as they are not your object of interest.

Author Response

Thank you for your comments on our paper. We have revised our paper based on your comments. Please refer to the attachment.

Reviewer 2 Report

1. It is a very good title, and it is easily understood by the rest of the paper

2. Rather than say "Achieved good performance in complex" in the abstract, it would be better if some statistical results were given instead of saying "Achieved good performance in complex".

3. I would appreciate it if you could also mention some words about the future work in your abstract

4. Reference [2] is used before [1,2], so please do not repeat it more than once.

5. I would like to introduce this line with a reference to "With the great success of deep learning in the fields of image classification and image segmentation, more and more scholars are focusing on the study of implementing visible image power line extraction based on deep learning"

6. There are also some references that need to be added to the paragraph starting from line 83

7.  I was unable to find any literature on the advantage (1) of using block processing strategy. Can you please provide some literature on this?

8.  There should be a change in the title of Table 1 to UAV

9. In Figure 2, the images look different, so please provide the same images, and the name of the figure should also be specific so that it is easier to understand.

10.  I found the methodology diagram to be very impressive

11.  In Figure 4, the name should be given a bit more detail

12. What is the purpose of Figure 6? It seems to be very complicated, so please make it as simple as possible for the researchers to use

13. The experimentation process was too lengthy, and it was unclear what the results were going to be. I would appreciate it if you could make it short and provide the results and discussion in a separate section

14. "Experimental results show that the performance of this method has been greatly improved over other classical methods such as the traditional Mask RCNN method. If possible, please provide some statistical comparison between these methods."

15. It would be helpful if you could provide some recent references as well

Author Response

(The authors gave the same response as above.)

Round 2

Reviewer 1 Report

Thank you for your explanations. However, your explanations and revisions do not resolve the weakness identified in my original review report:

1. The experiment based on a very small dataset (6 images only). 

2. No state-of-the art has been identified and no comparison with their results. Although similar works have been identified, no explanation of how your work can address their weakness.

I appreciate the problem you are tackling is challenging, but you have not shown you have made a significant contribution. I feel your contribution is OK for a conference paper, but not for a journal paper.

Author Response

Thank you very much for your comments, we have carefully made changes to the article based on your comments, please see the attachment.

Round 3

Reviewer 1 Report

Thank you for reviewing the paper. This revision is a good step to the right direction. However, you have still not compared with the state-of-the-art and testing on your own images. Both LSD and EDLines are old image-processing based methods, but not the state-of-the-art. You should test your detector on publicly available dataset and compare your results with those using the same dataset. One such publicly available powerline dataset and the performance of several state-of-the-art deep learning models on the dataset can be found here:

TTPLA: An Aerial-Image Dataset for Detection and Segmentation of Transmission Towers and Power Lines

https://arxiv.org/abs/2010.10032

Furthermore, your Introduction is still talking about your powerline detection can be used for diagnosis. Without the ability to detect powerline of different conditions and other power transmission components such as the towers and insulators, I am not convinced your detector can be used for diagnosis. You should revise your introduction and discuss how your detector can be used for more relevant application cases.

Author Response

Based on your comments we have made changes to the article, please see the attachment.
